# Acoustic Characteristics of Microcellular Foamed Ceramic Urethane

**DOI:** 10.3390/ma15062007

**Published:** 2022-03-08

**Authors:** Jin Hong, Sung Woon Cha

**Affiliations:** School of Mechanical Engineering, Yonsei University, 50, Yonsei-ro, Seodaemoon-gu, Seoul 03722, Korea; jin.hong@yonsei.ac.kr

**Keywords:** microcellular foam, ceramic, urethane, sound absorption, porous, resonance

## Abstract

Noise pollution critically degrades the quality of human life, and its effects are becoming more severe due to rapid population growth and the development of industry and transportation. Acoustic wave aggregation in the 30–8000 Hz band can have a negative impact on human health, especially following continuous exposure to low-frequency noise. This study investigates the acoustic performance of microcellular foams made of a mixture of brittle and soft materials and their potential use as absorption materials. It is common to use porous materials to improve acoustic properties. Specimens prepared by mixing ceramic and urethane were made into microcellular foamed ceramic urethane by a batch process using carbon dioxide. The specimens were expected to exhibit characteristics of porous sound-absorbing materials. After measuring the acoustic characteristics using an impedance tube, a significant sound-absorption coefficient at a specific frequency was noted, a characteristic of a resonance-type sound-absorbing material. However, the sound-absorption properties were generally worse than those before foaming. Differences based on the size, shape, and structure of the pores were also noted. It will be necessary to check the effects of cellular morphological differences on the absorption properties by controlling the variables of the microcellular foaming process in a future study.

## 1. Introduction

Urbanization and rapid industrial development have led to severe environmental and social problems associated with noise pollution. Some specialists suggest that noise can contribute to mental and physical health problems. In addition, with improvements in living conditions, people become more interested in a comfortable and quiet personal environment [1]. Various methods, such as installing sound barriers and changes in structural design, are applied to address these issues. In addition, the design, material, chemical composition, and manufacturing process of sound-absorption materials have improved, and research is ongoing to develop eco-friendly sound-absorption materials with good performance at a low cost [2].

Metal and concrete are used as building materials for soundproof walls, mainly because of their low cost. However, these are disadvantageous because they do not blend with the surrounding landscape, obscure visibility, and pose a threat to birds. In addition, in large cities, the number of apartments or multifamily buildings housing multiple households in one building is increasing compared to single households.

Various methods to reduce the sound output of noise sources and block or reduce noise by sound absorption and sound-absorption are implemented to create a quiet environment. The methods used to solve these problems include installing sound barriers, designing structures, or changing locations. In addition, the effects of changing the design, material, chemical composition, and manufacturing process of sound-absorption materials are the subject of this study.

Acoustic materials are divided into sound-absorbing and sound-insulating materials. There are three main types of sound-absorbing materials: porous, plate, and resonant. The primary function of sound-absorbing materials is to absorb sound energy to reduce resonance and prevent the outward spread of sound. The surface area is increased by making the cross-section uneven so that the vibrations are absorbed in multiple places. Sound-absorbing materials with a higher density perform better than those with a lower density. Sound-insulating materials block or reflect sound energy, preventing noise from escaping. Porous materials are mainly used to control noise in the mid-and high-frequency bands [3,4,5]. Their performance depends on the size, shape, density, and porosity of the cells in the material. In polymer resins, viscosity is also a factor; thus, the properties vary depending on the foaming conditions. In addition, the effectiveness of sound-insulation materials depends on the thickness of the material.

Ceramic is a typical brittle material used as a resonance-type sound-insulating material, and urethane is a typical ductile material often used as a porous sound-insulating material (Table 1) [6,7,8]. This study focuses on the changes in the acoustic performance of the materials through microcellular foaming. This method, in which two materials with extremely opposite characteristic impedances and propagation constants are synthesized, maximizes the acoustic performance. Microcellular foaming forms pores in the base material and creates microsized bubbles with different sound-insulation properties than those of single panels. We measured the sound absorption or insulation performance developed due to the air layer and determined the similarities and differences between the resonance and porous acoustic materials.

## 2. Materials and Methods

Sound-absorbing materials are classified into porous, plate, and resonance types. Ceramic and urethane are sound-absorption materials used in various applications. Ceramic is mainly used as a resonance type, and urethane is mainly used as a porous type.

The performance of a porous sound-absorbing material depends on the thickness and air layer behind the material; however, in general, the sound-absorption coefficient is high for medium and high frequencies and low for low frequencies. Many microsized cells form in the sound-absorbing material, and when sound is incident on the air-permeable and porous material, the sound waves are transmitted through the air in the holes and penetrate the material. Part of the incident sound-wave energy is absorbed by the air friction in the gaps present in the material. Sound attenuation increases with heat loss because of heat conduction between the air in the gap and walls of the material and the vibration of small fibers and aggregates constituting the pores. Even low frequencies can be absorbed if the surface area of the material is increased and porous sound-absorbing material is installed away from the substrate.

Resonance-type sound-absorbing materials have a sound-absorbing effect at a specific frequency. Drilling holes in a plate to form pores causes the plate to act as a resonator with a high sound-absorption effect for a specific frequency. The underlying principle involves energy loss when air vibrates violently owing to resonance [9,10,11,12,13]. The resonance frequency is determined by the volume of the air layer behind the hole and the resistance of the hole. If the gap between the plate and the wall is widened, the resonance shifts to the low-frequency region. If a porous material, such as glass wool, is inserted behind the plate with empty holes, the sound-absorption coefficient (SAC) can be significantly increased in a fairly wide range around the resonant frequency. 

### 2.1. Theory and Approaches

Sound waves are transmitted by the compression and relaxation of medium particles. Furthermore, they can propagate through any gaseous, liquid, or solid medium. When different media types are located in parallel, sound waves inside one medium can penetrate the other medium. Sound waves are generated by applying instantaneous or continuous vibrations to a medium. Simultaneously, a slight pressure change occurs in the medium, which moves away from the source owing to the elasticity of the medium. Sound waves can undergo diffraction, refraction, interference, reflection, transmission, and absorption, depending on the transmission path conditions.

As sound waves travel and encounter other interfering substances, some wave energy is reflected, while some are absorbed. Sound waves absorbed by the interfering material are converted into internal thermal energy, which is then dissipated. Moreover, some of the waves travel through to the other side of the interfering material.

The intrinsic properties of a sound-absorbing material include its characteristic impedance and propagation constant. Characteristic impedance refers to a specific value that indicates the resistance to sound-wave propagation through the sound-absorbing material. The propagation constant represents the attenuation and phase information of the sound waves propagating through the sound-absorbing material. When sound waves propagate through the atmosphere, the variables determining their properties are density and temperature. In addition, airflow and porosity are critical parameters for predicting the performance of sound-absorbing materials. Airflow refers to the resistance encountered when air passes through a sound-absorbing material, and porosity is a measure of the volume of air in the sound-absorbing material [14]. The performance of the ceramic urethane sound-absorbing material can be predicted using the above-mentioned factors.

Porous materials, such as polyurethane foam, are sound-absorbing materials used for passive noise control in various industries. In general, in the case of a rigid frame or limp-type sound-absorbing material, an equivalent fluid model is feasible, and the sound absorption coefficient can be predicted using Johnson–Champoux–Allard (JCA) model. For this prediction, five non-acoustic properties are required: porosity, airflow resistivity, tortuosity, and thermal and viscous characteristic length. Moreover, in the case of a sound-absorbing material with an elastic frame, it is possible to predict the sound-absorption rate using the Biot–Allard model. For this prediction, in addition to the previous five non-acoustic properties, the following three additional elastic properties are required: Young’s modulus, loss factor, and Poisson’s ratio [15,16].

According to the ASTM E1050-19, the relation between the transfer function and the surface impedance is as follows (Figure 1) [17]:(1)$$Z_{s}\;=jZ_{air}\;\frac{\sin\left[k\left(x-s\right)-H\sin\left(kx\right)\right]}{H\sin\left(kx\right)-\cos\left[k\left(x+s\right)\right]}$$
where $Z_{s}$ is surface impedance, *j* is the phase change of sound waves propagating in the sound-absorbing materials, k is the wavenumber that wavelength what frequency is to time period, and $Z_{air}$ is the characteristic impedance of air.

The surface impedance is obtained by measuring the transfer function, *H*, between two microphones, fixed on top of the sound-absorbing sample, where *x* is the distance from the specimen to the first microphone, and *s* is the distance between the microphones. Taking into account the flow inside cylinder, it is necessary to switch the position of microphones used for measurement to resolve measurement errors. This can be conducted by switching the cable connection positions of the microphones, CH 1 and CH 2, as depicted in Figure 1.

### 2.2. Material Preparations

In this study, a ceramic urethane sheet (MISUMI Group Inc., Seoul, Korea, Product code No. UTSCM) was cross-linked after mixing the ceramic powder with urethane made of polyester polyol.

The batch process (Figure 2), a foaming method developed by Martini [18], enables the simple production of microcellular foaming. This process can achieve better results at similar strength, fracture toughness, and insulation conditions while using less material compared to that in microcellular foaming processes. Microcellular foam fabrication varies drastically in cell generation, depending on the saturating gas type, gas pressure, solubility, foaming temperature, and foaming time [19,20,21].

As the pressure of the saturated gas increases during microcellular foaming, the cell density increases. Conversely, as the saturation temperature, Ts increases, the cell diameter increases, and the cell density decreases. When the temperature of the gas Tg is higher than Ts, the diameter of the cell increases. In addition, the cell size is affected by the viscosity of the polymer resin because cells can be easily formed when the saturation temperature is high and the viscosity is low.

The specimens were saturated with carbon dioxide (Samheung, Seoul, Korea; product grade no. CO_2_) at 5 MPa for 100 h in a high-pressure vessel equipped with an electric heater at 100 °C. In the foaming phase, an oil bath (Chang Shin Science Co., Seoul, Korea, Product No. C-WHT) and 99.50% glycerin was used to produce microcellular-sized cells. The cell morphology was controlled by the foaming time and temperature, 30 or 90 s at 150 °C or 170 °C (Table 2). An electronic densimeter (Alfa Mirage, Model No. MD-300S, Osaka, Japan) and electronic scale (OHAUS, Model no. AR2130, Nänikon, Switzerland) were used to measure the densities and weights of the specimens, respectively [22]. These were prepared in two sizes, 60 mm (Figure 3, Table 3, and Figure 4) and 30 mm (Figure 5, Table 4, and Figure 6) in diameter, both with a thickness of 20 mm.

### 2.3. Measurement of Acoustic Properties

Two methods are typically used for measuring acoustic characteristics: measuring a sound wave incident vertically using an impedance tube and measuring a sound wave incident in an arbitrary direction in a reverberation room [23].

The sound absorption coefficient of the microcellular foamed ceramic urethane was measured using an impedance tube. The experiment was conducted in high- and low-frequency bands in accordance with ASTM E1050-19 [24,25]. Two impedance tubes (BSWA. Technology Co., Ltd., Beijing, China, Impedance tube model No. SW 466) were prepared: a 30 mm diameter tube was used to measure the high-frequency region, and a 60 mm diameter tube was used to measure the low-frequency region (Figure 7).

After configuring the measurement software to set the desired tube size and sound using a laptop connected to the impedance tube, microphone channels 1 and 2 were calibrated to 94 dB using a separate calibrator. Then, white noise was input for approximately 10 min to correct the result using a signal generator. After configuring the conditions of specimen thickness, measurement frequency, temperature, relative humidity, air density, and sound velocity, a specimen was inserted, and measurements were performed. 

An input value of 110 dB was fed to the speaker on the opposite side of the specimen, and the distance between the two microphones was 15 mm. The sound was transmitted for 30 s and then automatically turned off. Subsequently, the positions of the two microphones were reversed, and another measurement was performed to complete one measurement cycle. After three measurements, the average value was calculated and used as the final result. The specimen was airtight and clamped. The temperature in the tube was 25 °C, the relative humidity was RH45 %, and the speed of sound was 340 m/s.

## 3. Results

Radio waves can carry additional information if their frequency is increased. They undergo little diffraction; however, they are easily reflected when obstacles are encountered and absorbed when it rains or snows. Conversely, if their frequency is low, the waves are more strongly diffracted; thus, they easily surpass the wall-to-wall limitation, are robust to natural phenomena such as snow and rain, and are more effectively transmitted. In general, the sound frequency that humans can hear ranges from 20 Hz to 20,000 Hz, and the voice frequency band is between 300 Hz and 3400 Hz. The measured data were expressed as a function of frequency, and the absorption coefficient is 0 for perfect reflection and 1 for perfect absorption.

### 3.1. Sound Absorption Coefficient at Low-Frequency

Low-frequency waves have relatively long wavelengths and are advantageous for long-distance propagation because of their strong diffraction and low attenuation in the medium. They are characterized by high, medium, and low-void permeability. In the low-frequency region, because noise and vibration are transmitted simultaneously, the discomfort caused by vibration is greater than that caused by high-frequency sound. The sound-absorption effect was higher than the sound insulation effect, and the sound insulation effect was higher than the vibration damping effect. It can be seen that the non-foamed material absorbs about 50% of the sound at about 650 Hz. After foaming, the sound-absorbing function disappeared in almost all samples. However, it was confirmed that the specimen, which was advantageously foamed at 150 °C for 90 s, showed a perfect sound-absorption effect in the same frequency band (Figure 8 and Table 5).

### 3.2. Sound Absorption Coefficient at High-Frequency

High-frequency waves are disadvantageous for long-distance propagation because of their relatively short wavelength and significant attenuation in the propagation medium. They undergo little diffraction and have good permeability through gaps; however, they are ineffective for vibration transmission. They are better mitigated by sound insulation than sound absorption and sound absorption by vibration damping and dust proofing (Figure 9 and Table 6). The auditory sensitivity to high-frequency waves is high, but the vibration sensitivity of the human body is low, and the auditory sensitivity is the highest at approximately 4000 Hz.

## 4. Discussion

Low-frequency noise exists in natural sounds, such as waves, thunder, and wind, as well as in many constructed environments, such as subways, blowers, buses, air conditioning, heating systems, and vacuum pumps. Low-frequency noise affects the circulation, respiration, nervous system, endocrine system, and sleep by causing drowsiness, nerve fatigue, and vomiting.

There are different types of high-frequency noise, but the most common is a long, intermittent, and high-frequency sound. It generally occurs in machinery, such as motors or construction equipment, and includes loud and roaring noises made by neighbors and occasional listening to loud music using earphones. Occasionally, tinnitus is encountered, which sounds like a cicada, grass insect, or a mechanical sound. High-frequency sounds are sensitive to psychological and physical changes, and symptoms such as dizziness, hearing loss, and pain may accompany tinnitus symptoms.

Noise pollution is one of the major factors reducing the quality of human life, and it is rapidly becoming severe due to rapid population growth and the rapid development of modern industry and transportation. Industrial and domestic equipment and vehicles, especially automobiles, are sources of structural and airborne noise. Sound wave aggregation in the 30–8000 Hz band can adversely affect human health in urban and industrial areas, especially following exposure to low-frequency noise.

The use of porous media as sound-absorbing and insulating materials is an effective method for reducing noise pollution. In particular, microporous-polymer foam with viscoelastic behavior and numerous pores is one of the best acoustic damping materials for eliminating noise. It can convert negative energy into frictional heat energy from the vibration of the plate or air, each of which has a characteristic sound-absorption performance. If a porous sound-absorbing material is added to the background air layer, the sound-absorption rate increases further, and there is little change in the vibration sound absorption frequency. The range of sound absorption is generally less than 200–300 Hz, and the heavier the plate, the thicker the air layer, and the lower the range. Both vibration- and resonance-sound absorption occur for a porous board, and a sound absorption effect is exhibited. At low frequencies, the propagation is isothermal because there is sufficient time for heat conduction. At significantly high frequencies, the available time in the heat diffusion cycle from the hot region to the cold region is short, and the propagation is adiabatic. In the intermediate frequency range, smaller wavelengths increase the temperature gradient and result in a maximum energy loss.

Ceramic, a resonant sound-absorbing material, reduces noise by vibrating the air in the material holes near the resonance frequency.

The perforated plate structure has multiple parallel configurations of a Helmholtz resonator. Therefore, a perforated plate is considered to be a resonant system consisting of a mass and a spring. When the incident wave frequency matches the system resonant frequency, the air within the perforated plate holes creates strong oscillating friction that enhances the absorption effect, forms an absorption peak, and converts the sound energy into thermal energy. If the frequency of the incident wave is significantly different from the resonant frequency, the absorption effect is reduced. According to the classical theory of perforated plate absorption, the main factors affecting the sound absorption of the perforated plate are the plate thickness, aperture, perforation rate, and pore depth.

The depth of the pores significantly affects the size of the resonant absorption peak and resonant frequency of the perforated plate. The acoustic resistance remained constant as the pore depth increased and the acoustic loss decreased. Therefore, in general, when the pore depth increases, the resonant frequency decreases, and the resonant frequency decreases [26,27,28,29].

In this study, the frequency corresponding to the highest sound-absorption coefficient (SAC) did not fluctuate because of foaming or changes in the foaming conditions. While the overall sound absorption performance was poor, the material exhibited significantly high SAC at a specific frequency. Different cell morphologies can be made by controlling various process conditions in the microcellular foaming process such as foaming temperature and time, gas saturation temperature, saturated gas type, saturation time, and a number of foaming operations. By studying these different specimens, better information to characterize the acoustic properties of the specific morphology can be obtained. A study to identify changes in the resonance frequency after foaming of materials or due to foaming will be undertaken shortly. 

## 5. Conclusions

This study investigated the acoustic performance of a microcellular foamed material mixed with brittle and soft materials and its potential as a sound-absorption material. Among the various methods used by most researchers to improve acoustic properties, pores are the most commonly used, and they exhibit a significant improvement in sound- insulation properties. Thus, numerous microsized pores were created in the microcellular foamed ceramic urethane fabricated using a batch process. Accordingly, it was expected to exhibit the characteristics of a porous sound-absorbing material. As a result of measuring the sound-absorption coefficient according to ASTM E-1090 in the low- and high-frequency bands using an impedance tube, it was determined that the resonance-type absorbent material possesses a significant sound-absorption coefficient at a specific frequency. This result suggests that despite the formation of numerous pores through microcellular foaming in urethane, the sound-absorption properties are lower than those before foaming. However, it was concluded that the ceramic urethane, which was microcellular foamed at 150 °C for 30 s, almost completely blocked the noise at a specific frequency, even at a thickness of approximately 20 mm.

## Figures and Tables

**Figure 1 materials-15-02007-f001:**
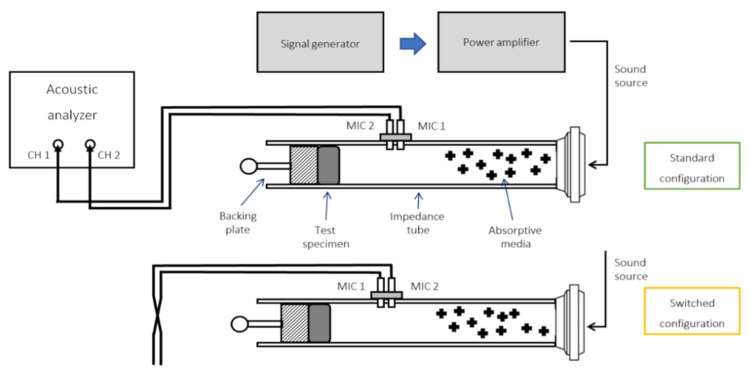
Schematic of the instrumentation used in this study to measure the sound absorption coefficient.

**Figure 2 materials-15-02007-f002:**
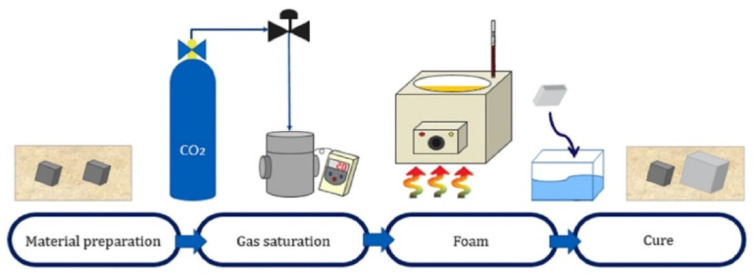
The procedure of the batch process.

**Figure 3 materials-15-02007-f003:**
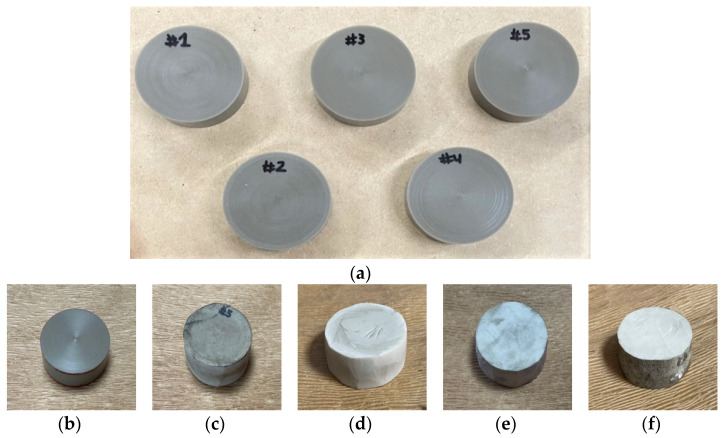
Specimens for low-frequency measurements: (**a**) Non-foamed; (**b**) Original; (**c**) Microcellular foamed at 150 °C and 30 s; (**d**) Microcellular foamed at 150 °C and 90 s; (**e**) Microcellular foamed at 170 °C and 30 s; (**f**) Microcellular foamed at 170 °C and 90 s.

**Figure 4 materials-15-02007-f004:**
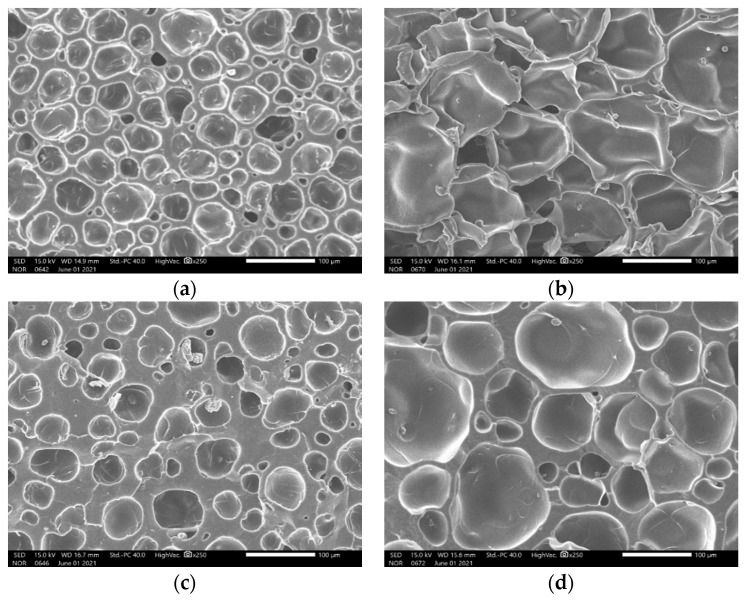
Scanning electron micrograph of ceramic urethane specimens for low-frequency measurements: (**a**) Microcellular foamed at 150 °C and 30 s (at a magnification of ×250); (**b**) Microcellular foamed at 150 °C and 90 s (at a magnification of ×250); (**c**) Microcellular foamed at 170 °C and 30 s (at a magnification of ×250); (**d**) Microcellular foamed at 170 °C and 90 s (at a magnification of ×250).

**Figure 5 materials-15-02007-f005:**
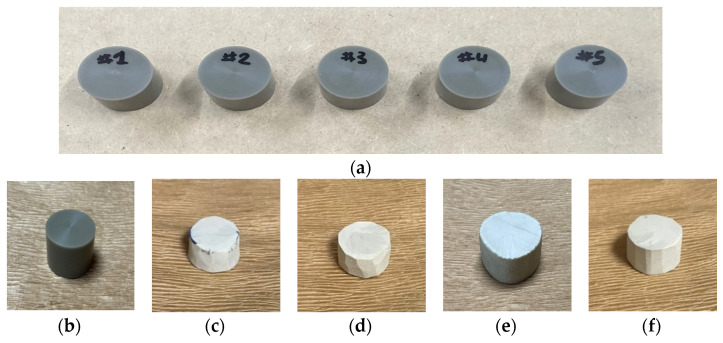
Specimens for high-frequency measurements: (**a**) Non-foamed; (**b**) Original; (**c**) Microcellular foamed at 150 °C and 30 s; (**d**) Microcellular foamed at 150 °C and 90 s; (**e**) Microcellular foamed at 170 °C and 30 s; (**f**) Microcellular foamed at 170 °C and 90 s.

**Figure 6 materials-15-02007-f006:**
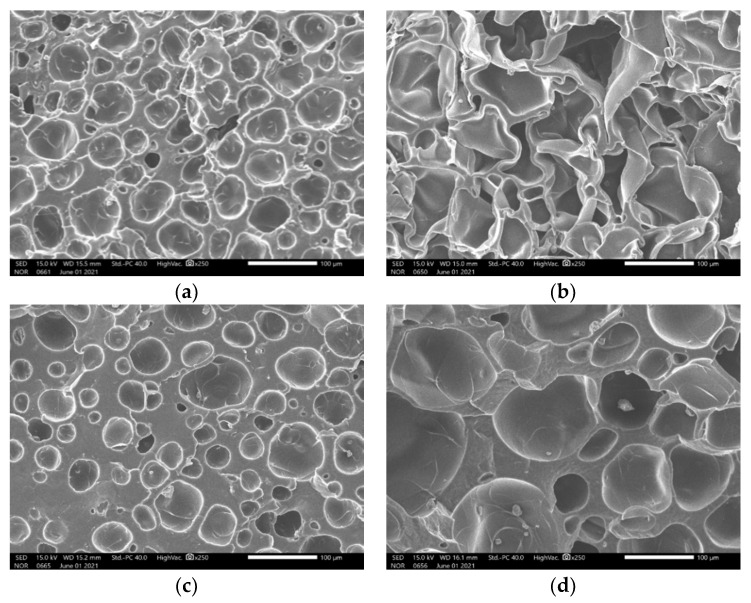
Scanning electron micrograph of ceramic urethane specimens for high-frequency measurements: (**a**) Microcellular foamed at 150 °C and 30 s (at a magnification of ×250); (**b**) Microcellular foamed at 150 °C and 90 s (at a magnification of ×250); (**c**) Microcellular foamed at 170 °C and 30 s (at a magnification of ×250); (**d**) Microcellular foamed at 170 °C and 90 s (at a magnification of ×250).

**Figure 7 materials-15-02007-f007:**
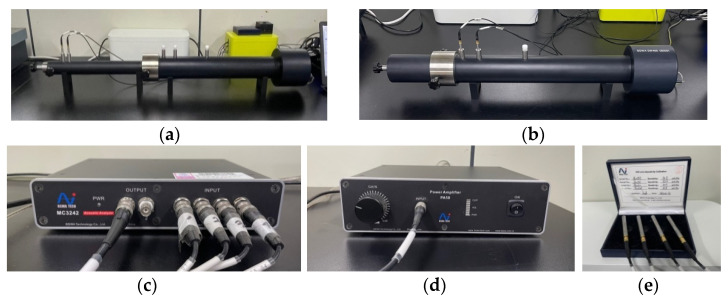
Measuring equipment: (**a**) Impedance tube to measure sound absorption coefficient −30 mm; (**b**) Impedance tube to measure sound absorption coefficient −60 mm; (**c**) Acoustic analyzer; (**d**) Power amplifier; (**e**) Microphones.

**Figure 8 materials-15-02007-f008:**
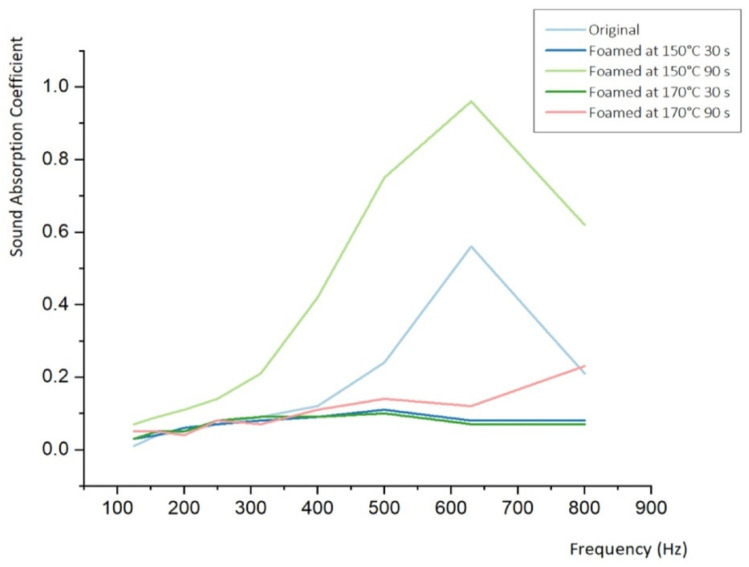
Sound absorption coefficient–frequency curve of microcellular foamed ceramic urethane obtained using an impedance tube (Low-frequency).

**Figure 9 materials-15-02007-f009:**
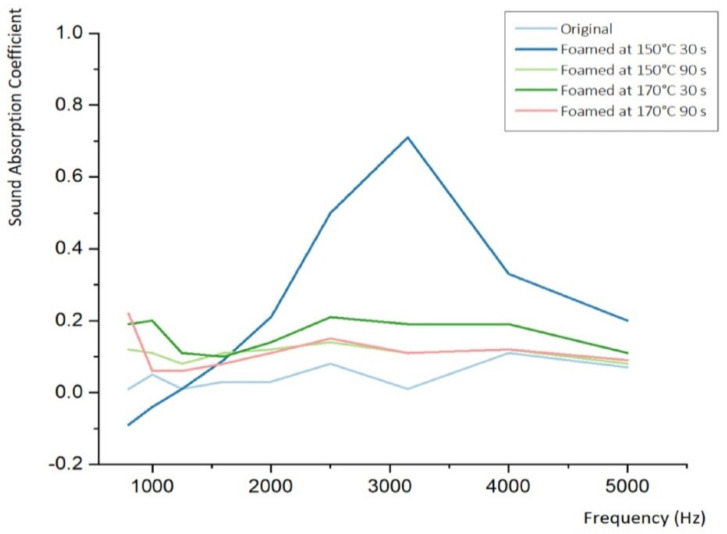
Sound absorption coefficient–frequency curve of microcellular foamed ceramic urethane obtained using an impedance tube (high-frequency).

**Table 1 materials-15-02007-t001:** Material properties.

Property	Ceramic	Urethane
Softening/melting point (K)	2323	365
Yield strength (MPa)	5000	26–31
Tensile strength (MPa)	250–550	58
Young’s modulus (GPa)	385–392	2.6–3
Stiffness (GPa)	100	2
Ductility (%EL)	-	2–5.5
Specific heat (J/g°C)	0.8	1.9
Thermal conductivity (W/m K)	16–29	0.12–0.18
Ionization energy (kJ/mol^−1^ of O_2_)	Large and Positive	≈−400

**Table 2 materials-15-02007-t002:** Experimental conditions.

Microcellular Foam—Batch Process	
Saturation gas	Carbon dioxide
Saturation pressure (MPa)	5
Saturation time (Hours)	100
Saturation temperature (°C)	100
Foaming fluid	Glycerin
Foaming time (s)	30/90
Foaming temperature (°C)	150/170
Experimental temperature (°C)	20 ± 3
Humidity (RH %)	65

**Table 3 materials-15-02007-t003:** Specifications of microcellular foamed ceramic urethane for low-frequency measurements.

Category	Original	Foamed at 150 °C 30 s	Foamed at 150 °C 90 s	Foamed at 170 °C 30 s	Foamed at 170 °C 90 s
Weight (g)	10.054	10.077	10.641	10.111	11.050
Density (g/cm^3^)	1.259	0.859	0.412	0.690	0.461
Solubility (%)	-	7.14	6.96	7.37	7.06
Void fraction (%)	-	31.61	67.30	45.19	63.41
Ave. of cell size (μm)	-	28.350	52.129	28.267	56.487

**Table 4 materials-15-02007-t004:** Specifications of microcellular foamed ceramic urethane for high-frequency measurements.

Category	Original	Foamed at 150 °C 30 s	Foamed at 150 °C 90 s	Foamed at 170 °C 30 s	Foamed at 170 °C 90 s
Weight (g)	2.14	1.01	0.85	1.01	1.06
Density (g/cm^3^)	1.256	0.688	0.471	0.663	0.512
Solubility (%)	-	12.52	6.32	12.72	6.25
Void fraction (%)	-	45.4	62.6	47.4	61.4
Ave. of cell size (μm)	-	31.251	44.883	30.640	63.972

**Table 5 materials-15-02007-t005:** Results of SAC for measuring at low-frequency.

Frequency	Original	Foamed at 150 °C 30 s	Foamed at 150 °C 90 s	Foamed at 170 °C 30 s	Foamed at 170 °C 90 s
125	0.01	0.03	0.07	0.03	0.05
160	0.04	0.04	0.09	0.05	0.05
200	0.05	0.06	0.11	0.05	0.04
250	0.07	0.07	0.14	0.08	0.08
315	0.09	0.08	0.21	0.09	0.07
400	0.12	0.09	0.42	0.09	0.11
500	0.24	0.11	0.75	0.10	0.14
630	0.56	0.08	0.96	0.07	0.12
800	0.21	0.08	0.62	0.07	0.23

**Table 6 materials-15-02007-t006:** Results of SAC for measuring at high-frequency.

Frequency	Original	Foamed at 150 °C 30 s	Foamed at 150 °C 90 s	Foamed at 170 °C 30 s	Foamed at 170 °C 90 s
800	0.01	−0.09	0.12	0.19	0.22
1000	0.05	−0.04	0.11	0.20	0.06
1250	0.01	0.01	0.08	0.11	0.06
1600	0.03	0.09	0.11	0.10	0.08
2000	0.03	0.21	0.12	0.14	0.11
2500	0.08	0.50	0.14	0.21	0.15
3150	0.01	0.71	0.11	0.19	0.11
4000	0.11	0.33	0.12	0.19	0.12
5000	0.07	0.20	0.08	0.11	0.09

## Data Availability

Not applicable.

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
