# Peer review of "Acoustic Characteristics of Microcellular Foamed Ceramic Urethane"

_materials, 2022, doi:10.3390/ma15062007_

Round 1

Reviewer 1 Report

1. In introduction “During the COVID-19 pandemic, working and studying from home have significantly increased, and 38 noise between neighbors has become a serious problem. ” I consider that there is no necessary connection between noise and COVID-19. 2. In Figure 8, It seems that there are not enough data points. It is recommended to add more. I suggest authors would not adopt center frequency of one third octave, because this test is for small sample. Basicly, we adopt center frequency of one third octave for big sample in reverberation room. Unless the author gives a good reason in text. 3. In conclusion, author used “soundproofing”. I suggest authors should use sound-absorption 4. Authors maybe lose some important citation in the introduction.[1] Design, fabrication and sound absorption test of composite porous matamaterial with embedding I-plates into porous polyurethane[J], Applied Acoustics, 2021, 175, 107845. This paper is all about the broadband sound absorption design of porous materials.

Author Response

Dear: Reviewer (1)

Thank you very much for reviewing our paper. We have attempted to answer the questions posed and revised the manuscript to incorporate the comments by the reviewer. We have also improved the quality of all figures in the manuscript. The revised text is marked in red color while comments are marked in blue color in the revised manuscript.

This manuscript presents the investigations of the changes in the acoustic characteristics of ceramic urethanes manufactured using the microcellular foaming process (MCP). Additionally, a comparison with the characteristics of existing sound absorption materials, such as resonance and porous type materials, suggests new singularities. However, my consideration about of the manuscript must be addressed based on my comments.

[Q-1] In introduction “During the COVID-19 pandemic, working and studying from home have significantly increased, and 38 noise between neighbors has become a serious problem.” I consider that there is no necessary connection between noise and COVID-19.

(Author): Thank you very much for your helpful comments. We agree with your opinion that there is no direct relation between noise-induced problems in human life and the pandemic. The introduction has been modified to exclude the contents related to the COVID-19 pandemic. Please check lines 40-42.

[Q-2] In Figure 8, It seems that there are not enough data points. It is recommended to add more. I suggest authors would not adopt center frequency of one third octave, because this test is for small sample. Basically, we adopt center frequency of one third octave for big sample in reverberation room. Unless the author gives a good reason in text.

(Author): Thank you so much for your comments. Generally, two standards are used to measure the sound absorption coefficient. One is ASTM C423 (Standard test method for sound absorption and sound absorption coefficients by the reverberation room method) that you suggested, and the other is ASTM E1050 (Standard test method for impedance and absorption of acoustical materials using a tube, two microphones, and a digital frequency analysis system). Both methods are recognized as international standards, and the measurement frequency band is almost the same, approximately 100-5000Hz.

The purpose of our article is not to manufacture sound-absorbing materials but to explain the sound-absorbing properties shown by microcellular foamed ceramic urethane. Therefore, rather than the validity of the experimental method, we chose to focus on the new results revealed through the experiment and the reason for this phenomenon. This will be proven with the results derived from the experiment using the impedance tube and tested again in a new way with a newly manufactured specimen.

[Q-3] In conclusion, author used “soundproofing”. I suggest authors should use sound-absorption.

(Author): Thank you very much for your helpful comments. Based upon your recommendation, we looked up the clear definition of soundproofing and sound absorption.

  • Soundproofing is defined as the process of blocking noise from either entering or exiting a room. You can do this through various methods and can block out a range of sounds.
  • Sound absorption, however, is the process of absorbing sound waves within a room and is also known as acoustic treatment. This process involves reducing or eliminating echo, reverberation, and amplification. In short, sound absorption improves sound quality within a space rather than preventing sound transfer.

Consequently, “sound absorption” seems to be the more suitable expression to describe the acoustic properties of microcellular foamed ceramic urethane material. Please check lines 19, 20, 33, 34, 44, 48, 74, 258, 269, 341 and 351.

[Q-4] Authors maybe lose some important citation in the introduction. [1] Design, fabrication and sound absorption test of composite porous material with embedding I-plates into porous polyurethane[J], Applied Acoustics, 2021, 175, 107845. This paper is all about the broadband sound absorption design of porous materials.

(Author): Thank you very much for recommending the perfect article. Your recommendations have been a great help in enhancing our knowledge. The manuscript has been edited to include the citation. Please check lines 129 and 399-400.

Reviewer 2 Report

The present work investigates the acoustic performance of microcellular foams made of a mixture of brittle and soft materials, and performed a significant sound-absorption coefficient at a specific frequency. The results suggest that the resonance type absorbent material possesses a significant sound-absorption coefficient and the soundproofing properties are lower than those before foaming. Overall, the performance sounds good and a series of results and discussion are also reasonable. However, some issues should be addressed.

1, authors should figure out the significance and real-life application in the Abstract section.

2, As far as I am concerned, it is just to meet the test conditions of acoustic properties, and how to achieve the strength requirements in practical application, please give more details.

Author Response

Dear: Reviewer (2) 

Thank you very much for reviewing our paper. We have attempted to answer the questions posed and revised the manuscript to incorporate the comments by the reviewer. We have also improved the quality of all figures in the manuscript. The revised text is marked in red color while comments are marked in blue color in the revised manuscript. 

This manuscript presents the investigations of the changes in the acoustic characteristics of ceramic urethanes manufactured using the microcellular foaming process (MCP). Additionally, a comparison with the characteristics of existing sound absorption materials, such as resonance and porous type materials, suggests new singularities. However, my consideration about of the manuscript must be addressed based on my comments. 

[Q-1] Authors should figure out the significance and real-life application in the Abstract section. 
(Author): Thank you very much for your helpful comments. In order to be faithful to the content of the article, it was not possible to insert the contents of actual application cases. In view of this, directions for further study to improve the deficiencies have been added to the abstract. 
(Revised text) 
‘Abstract’ 
Noise pollution critically degrades the quality of human life and its effects are becoming more severe due to rapid population growth and the development of industry and transportation. Acoustic wave aggregation in the 30–8000 Hz band can have a negative impact on human health, especially following continuous exposure to low-frequency noise. This study investigates the acoustic performance of microcellular foams made of a mixture of brittle and soft materials and their potential use as absorption materials. It is common to use porous materials to improve acoustic properties. Specimens prepared by mixing ceramic and urethane were made into microcellular foamed ceramic urethane by a batch process using carbon dioxide. The specimens were expected to exhibit characteristics of porous sound-absorbing materials. After measuring the acoustic characteristics using an impedance tube, a significant sound-absorption coefficient at a specific frequency was noted;, a characteristic of a resonance-type sound-absorbing material. However, the sound-absorption properties were generally worse than those before foaming. Differences based on the size, shape, and structure of the pores were also noted. It will be necessary to check the effects of cellular morphological differences on the absorption properties by controlling the variables of the microcellular foaming process in a future study.

[Q-2] As far as I am concerned, it is just to meet the test conditions of acoustic 
properties, and how to achieve the strength requirements in practical application, please give more details. 
(Author): Thank you very much for your helpful comments. Essentially, my primary purpose in this article is to present the results of the simultaneous appearance of two types of sound absorption properties — the resonance type and the porous type — in the microcellular foamed ceramic urethane sound absorption material. Therefore, we think that a little more in-depth research was needed before devising the usage of the material, the applicable part, or the applicable method. Nevertheless, I think your opinion is very valid. Therefore, we added a supplementary explanation about the possibility of changing the state and properties by microcellular foaming process (MCP) to the discussion part of the paper. Please check the lines 331-338. 
(Revised text) 
In this study, the frequency corresponding to the highest sound-absorption coefficient (SAC) did not fluctuate because of foaming or by changes in the foaming conditions. While the overall sound absorption performance was poor, the material exhibited significantly high SAC at a specific frequency. Different cell morphologies can be made by controlling various process conditions in the microcellular foaming process like foaming temperature and time, gas saturation temperature, saturated gas type, saturation time, and number of foaming operations. By studying these different specimens better information to characterize the acoustic properties of the specific morphology can be obtained. A study to identify changes in the resonance frequency after foaming of materials or due to foaming will be undertaken shortly. 

Reviewer 3 Report

Dear authors, dear editor,

The paper I received for reviewing investigates the acoustic absorbing properties of Urethane-ceramic foams.

The experimental design of the study is extremely poor and the study consists of 5 foams tested against low and high frequencies. The study does not allow to say much, because commercial references are missing and the only comparison is on the urethane-ceramic foams production process. The author also made a intrinsic characterization but it was not discussed in the paper (except reporting table and SEM images) and it was not combined to findings of the acoustic tests. In other words: In this way this is just a technical report because the reasons for the results registered are not investigated.

The article reports a lot of acoustic theories which do not deal with the study itself, producing excess of theoretic background which won’t be needed afterwards (because results are not discussed indeed).

Other criticisms:

Materials and methods:

  • Here the data should be concise reported without need for such a long introduction
  • The figure 1 is not introduced, and it is not immediate to understand which are the difference between the two configurations.

Results and discussion:

  • The text contains still part of the template!
  • Graphic and tables in this section are simply placed, without any presentation of the results nor discussion.

  • Figure 8 and 9 would suggest that the sound absorption coefficient is low for every foam except 150° for 90 for low frequencies and 150° for 30 for high frequencies… Overall it does not seem a good result.

  • Table 5 and 6 are the same than the figure 8 and 9. Please keep only one.

  • The discussion reported has almost nothing to do with the study, the only moment in which the authors try to partially discuss their results is in the conclusion part.

Author Response

Dear: Reviewer (3) 

Thank you very much for reviewing our paper. We have attempted to answer the questions posed and revised the manuscript to incorporate the comments by the reviewer. We have also improved the quality of all figures in the manuscript. The revised text is marked in red color while comments are marked in blue color in the revised manuscript. 

This manuscript presents the investigations of the changes in the acoustic characteristics of ceramic urethanes manufactured using the microcellular foaming process (MCP). Additionally, a comparison with the characteristics of existing sound absorption materials, such as resonance and porous type materials, suggests new singularities. However, my consideration about of the manuscript must be addressed based on my comments. 

[Q-1] The experimental design of the study is extremely poor and the study consists of 5 foams tested against low and high frequencies. The study does not allow to say much, because commercial references are missing and the only comparison is on the urethane-ceramic foams production process. The author also made a intrinsic characterization but it was not discussed in the paper (except reporting table and SEM images) and it was not combined to findings of the acoustic tests. In other words: In this way this is just a technical report because the reasons for the results registered are not investigated. 

(Author): Thank you very much for your helpful comments. We have performed various other experiments in addition to the experiments described in the article.  

In the case of acoustic characteristics experiments, After foaming plate-shaped specimens and cutting them to fit inside the impedance tube, they were stacked on top of each other according to the experimental plan, and the same test was performed. Although the stacks look unstable in the photo, there is no possibility of about sound leakage because it is a very tight fit when installed in the impedance tube. 

If we include the results of the above experiments in the manuscript, basic information about the data must also be included. In addition, a cross-check is required to provide authenticated data. We adjudged that the above data did not fit the objective of this journal, and so only the necessary parts were included. We have compiled only the data that we can convey clearly to avoid confusing the readers. If you think that the revised manuscript is still insufficient, please specify the part you think is necessary and let us know again so that we can provide additional data.  

[Q-2] The article reports a lot of acoustic theories which do not deal with the study itself, producing excess of theoretic background which won’t be needed afterwards (because results are not discussed indeed). 

(Author): Thank you very much for your helpful comments. As you are well aware, urethane and ceramics are widely used for noise reduction. However, the properties of the two materials are completely different. Although both are sound-absorbing materials, urethane is a porous type and ceramic is a resonance type material; hence, the method for sound absorption is completely different in theory. In addition, the microcellular foaming process was expected to maximize sound absorption or soundproofing effect, but this study showed the opposite result. Therefore, detailed theoretical backgrounds and explanations were given to inform the readers that the results of this experiment are different from the experimental results of standard urethane and ceramic sound-absorbing materials. 

[Q-3] Here the data should be concise reported without need for such a long introduction. The figure 1 is not introduced, and it is not immediate to understand which are the difference between the two configurations. 

(Author): Thank you very much for your helpful comments.  

Information to be conveyed in Figure 1: The difference between the two configurations is that the orientation direction used for the microphones is changed for the measurements. This can be achieved by switching the cable connection position of the microphones, CH 1 and CH 2. Additional reference material has been added to help with comprehension. Please check lines 133-136. 

(Revised text) 

The surface impedance is obtained by measuring the transfer function, H, between two microphones, fixed on top of the sound-absorbing sample, where x is the distance from the specimen to the first microphone, and s is the distance between the micro-phones. Taking into account the flow inside the cylinder, it is necessary to switch the position of the microphones, used for measurement, to resolve measurement errors. This can be done by switching the cable connection positions of the microphones, CH1 and CH2, as depicted in Figure 1. And more details, refer ASTM E1050-19. The relation between the transfer function and the surface impedance is as follows (Figure 1) [17]: 

[Q-4] The text contains still part of the template! Graphic and tables in this section are simply placed, without any presentation of the results nor discussion. Figure 8 and 9 would suggest that the sound absorption coefficient is low for every foam except 150° for 90 for low frequencies and 150° for 30 for high frequencies… Overall it does not seem a good result. 

(Author): Thank you for your kind comments. You have given a very accurate picture of the results. Since the replicable in multiple tests (at least 3 to 5 times) according to ASTM E1050-19, we have concluded that this result is meaningful and valuable. Since the experimental results cannot be modified to report good results, we will keep the continue studying this phenomenon more deeply. Thanks for the excellent checking. 

[Q-5] Table 5 and 6 are the same than the figure 8 and 9. Please keep only one. 

(Author): Thank you so much for your comments. As you understood, Table 5 - Figure 8 and Table 6 - Figure 9 are the results of the same tests. These values have been tabulated as well as represented visually to help the readers understand this article more easily and quickly. So, we have a plan to keep all the data. Nevertheless, if you feel that the duplication of data is excessive, it would be a great help to us if you could give us your opinion on which data you think is more helpful to readers. 

[Q-6] The discussion reported has almost nothing to do with the study, the only moment in which the authors try to partially discuss their results is in the conclusion part. 

(Author): Thank you very much for your helpful comments. In this article, our aim was to present the effects of the simultaneous presence of two types of sound-absorbing properties, a resonance type, and a porous type, in the microporous foamed ceramic urethane sound-absorbing material. Since it is a new approach, explanations of materials, manufacturing methods, foaming methods, and measurement methods take up a large part of the manuscript. In-depth research related to materials is needed to evaluate the suitability of the material’s use, application area, and application method. Nevertheless, I think your opinion is very valid. Therefore, a supplementary explanation about the possibility of changing the state and properties of the material by microcellular foaming process (MCP) has been added to the discussion part of the paper. Please check the lines 331-338. 

(Revised text) 

Different cell morphologies can be made by controlling various process conditions in the microcellular foaming process like foaming temperature and time, gas saturation temperature, saturated gas type, saturation time, and number of foaming operations. By studying these different specimens better information to characterize the acoustic properties of the specific morphology can be obtained. A study to identify changes in the resonance frequency after foaming of materials or due to foaming will be undertaken shortly. 

Reviewer 4 Report

In this work, the authors investigated the acoustic performance of a microcellular foamed material mixed with brittle and soft materials, namely ceramic urethane. The obtained microcellular foamed at 150 °C for 30 s, almost completely blocked the  noise at a specific frequency, even at a thickness of approximately 20 mm.

The conclusions are justified by the proposed approach and experiments, and corrections were included within the last uploaded manuscript, therefore I recommend the paper publication in its present form.

Author Response

Dear: Reviewer (4)

Thank you very much for reviewing our paper. This manuscript presents the investigations of the changes in the acoustic characteristics of ceramic urethanes manufactured using the microcellular foaming process (MCP). Additionally, a comparison with the characteristics of existing sound absorption materials, such as resonance and porous type materials, suggests new singularities.

[Q] In this work, the authors investigated the acoustic performance of a microcellular foamed material mixed with brittle and soft materials, namely ceramic urethane. The obtained microcellular foamed at 150 °C for 30 s, almost completely blocked the noise at a specific frequency, even at a thickness of approximately 20 mm.

The conclusions are justified by the proposed approach and experiments, and corrections were included within the last uploaded manuscript, therefore I recommend the paper publication in its present form.

(Author): Thank you very much for your helpful comments. We would like to thank you for your positive evaluation of the results of our study. We agree with your opinion and plan to derive new characteristics, singularities, and phenomena through continuous investigation and research.

Round 2

Reviewer 3 Report

Dear editor, dear authors,

I am surprised how an article that I firmly rejected was corrected in less than 1 month. Indeed, this technical report is only slightly modified and therefore I confirm the rejection for the simple reasons that there is no understanding on why the results found are these. Morphology of the foams was not explained and not coupled with the test results.

Author Response

Dear: Reviewer (3)

Thank you again for second reviewing our paper. We already told you that this manuscript presents the investigations of the changes in the acoustic characteristics of ceramic urethanes manufactured using the microcellular foaming process (MCP). Additionally, a comparison with the characteristics of existing sound absorption materials, such as resonance and porous type materials, suggests new singularities.

[Q] I am surprised how an article that I firmly rejected was corrected in less than 1 month. Indeed, this technical report is only slightly modified and therefore I confirm the rejection for the simple reasons that there is no understanding on why the results found are these. Morphology of the foams was not explained and not coupled with the test results.

(Author): We were given only ten days to make changes, so we tried to improve it using the best method possible during those ten days. Therefore, it was difficult to change the overall content of the study, and we ask for your understanding that we have taken into consideration the part where the essential content of our study may change if the overall content is changed.

As for the morphology of the microcellular foamed specimens, the size, density, void fraction, distribution, shape, and morphology of the cell can be confirmed through SEM images. (Please refer to figures 5 and 6.) Furthermore, the test results that we discovered in this study are a phenomenon that appears at a specific frequency rather than patterning, and that means it can not be any matching with morphology to test results. Although our answer is lacking, we sincerely hope that it has been of some help to your understanding.
